# Extraction and Recovery of Critical Metals from Electronic Waste Using ISASMELT™ Technology

Stuart Nicol *, Benjamin Hogg, Oscar Mendoza and Stanko Nikolic

Glencore Technology, 180 Ann St, Brisbane 4000, Australia
* Correspondence: stuart.nicol1@glencore.com.au

**Abstract:** Electronic goods are a major consumer of many critical metals, including copper, nickel, tin, zinc, lead, and precious metals. The processing of end-of-life electronic equipment (E-Scrap) is becoming increasingly important to maintain the supply of the critical metals required globally, and to reduce environmental pollution. Currently, the dominant route for E-Scrap processing is pyrometallurgical processing, with the first stage of processing being reductive smelting to produce a black copper and a 'clean' discard slag. The management of the slag in this first step is central to the success of the E-Scrap recycling process. The E-Scrap ISASMELT™ furnace has a highly turbulent bath, providing conditions that generate high rates of zinc fuming and allow a wide range of operable slag conditions. This enables efficient E-Scrap smelting to occur, whilst overcoming the challenges associated with alternative technologies. Operable slag compositions and high zinc fuming are heavily influenced by kinetic processes, with piloting critical to understanding the performance of this process. ISASMELT™ pilot tests were performed, with a wide range of fluxing targets tested to confirm these benefits. The testing demonstrated that high levels of zinc fuming (>80%) are obtained in the E-Scrap ISASMELT™ furnace, decreasing the iron and silica flux additions required to manage the detrimental viscosity effects of zinc in the slag. In addition, it was demonstrated that slags containing high concentrations of alumina (>10 wt%) are operable in an ISASMELT™ furnace. The ISASMELT™ technology was demonstrated to be the only E-Scrap furnace technology able to produce a 'clean' discard slag with low concentrations of zinc and minimal fluxing requirements.

**Keywords:** ISASMELT™; E-Scrap; E-Waste; recycling; ISACYCLE™; critical metals; fuming





## 1. Introduction

Critical metals are important for modern societies, with many of them used in a broad range of applications. One of these applications is electronics, both consumer and non-consumer electronics. The critical metals in electronics include copper, nickel, tin, zinc, lead, and precious metals. Currently 17% of tin [1], 21% of silver [2] and 7% of gold [3] produced annually are used in electronics. To move towards a sustainable society and circular economy, end-of-life electronics (E-Scrap) need to be responsibly processed to recover these critical metals. Reviews of E-Scrap have been performed, outlining the processes used and challenges encountered when smelting this material [4,5].

The composition of E-Scrap varies significantly, dependent on the electronic good, the manufacturer, and the year of manufacture. A table showing some E-Scrap compositions reported in the literature, is given in Table 1. Due to the high concentration of copper and the lower concentration of tin, nickel, and precious metals, E-Scrap is usually treated pyrometallurgically, using the secondary copper processing route. However, there are two major challenges when smelting E-Scrap—the formation of operable slags and the management of zinc. These challenges occur due to the high concentrations of aluminium in the feed and the low rates of zinc fuming reported for alternative furnace technologies [6,7]. Both the alumina and the zinc in slag stabilise the spinel phase, which increases the slag

liquidus temperature and results in higher operating temperatures, a higher slag viscosity, or both [8–10].

**Table 1.** Survey of E-Scrap Composition.

|  | Ref. 1 [11] | Ref. 2 [12] | Ref. 3 [13] | Ref. 4 [14] |
|---|---|---|---|---|
| Cu (wt%) | 20 | 15.6 | 22 | 6.9 |
| Al (wt%) | 2 | - | - | 14.2 |
| Pb (wt%) | 2 | 1.35 | 1.55 | 6.3 |
| Zn (wt%) | 1 | 0.16 | - | 2.2 |
| Ni (wt%) | 2 | 0.28 | 0.32 | 0.85 |
| Fe (wt%) | 8 | 1.4 | 3.6 | 20.5 |
| Sn (wt%) | 4 | 3.24 | 2.6 | 1 |
| Sb (wt%) | 0.4 | - | - | 20 |
| Au (ppm) | 1000 | 420 | 350 | 20 |
| Ag (ppm) | 2000 | 1240 | - | 200 |
| $SiO_2$ (wt%) | 15 | 41.86 | - | - |
| $Al_2O_3$ (wt%) | 6 | 6.97 | - | - |
| Alkali and Alkaline Earth Oxides (wt%) | 6 | 9.95 | - | - |
| Titanates and mica (wt%) | 3 | - | - | - |

ISASMELT™ technology is well established as one of the leading technologies for secondary copper smelting. The ISASMELT™ Technology was developed at Mount Isa Mines in Queensland, Australia, through rigorous lab testing, pilot testing, and demonstration-scale plant trials. The ISASMELT™ furnace is a bath smelting technology, with the ISASMELT™ lance submerged in the slag layer and injecting reaction gases directly into the molten bath. In this furnace, the slag phase is used to carry oxygen to the feed materials that are digested in the bath. This resulted in the first commercial ISASMELT™ plant being constructed in 1989 [15]. The technology was initially developed to process primary copper concentrates and primary lead concentrates, but it was quickly identified that the technology was able to be used for smelting and/or converting most primary and secondary base metal feed materials [16]. The leading E-Scrap recyclers in Europe, Aurubis Lunen and Umicore Hoboken, both use ISASMELT™ Technology to recover critical metals from secondary copper feed materials—and have done for more than 20 years [16]. The ISASMELT™ Technology is adaptable and suited to both continuous and batch processing of secondary base metal feeds, with the batch processing route enabling both smelting and converting to occur in a single vessel.

The challenges when processing E-Scrap will be reviewed and the adaption of ISASMELT™ technology to successfully address these challenges will be explored. The secondary copper process chemistry and slag design will be reviewed to understand E-Scrap Smelting. The pilot results for an E-Scrap ISASMELT™ furnace will be discussed and examined.

## 2. The Chemistry of Secondary Copper Smelting (Low-Grade Scrap)

Secondary copper from post-consumer waste and other urban waste streams differs from primary copper in that the material is low in sulphur. As such, a different smelting chemistry is used. For high and medium grade copper scraps, a simple 'melting' process can be used as the scrap is low in impurities [17].

When smelting low grade secondary copper and E-Scrap, three smelting steps are used to remove the impurities in the feed: reduction, oxidation, and fire refining [17]. The first

stage of smelting secondary copper materials is reductive smelting, which produces a black copper (rich in base metals) and a discard slag (preferably with very low concentrations of base metals). The second stage of smelting secondary copper materials is converting, where the black copper is oxidised to produce raw copper and side streams, both slags and dusts, that are rich in base metals. The raw copper can be processed in conventional Anode Furnaces to produce Anode Copper, which is electro-refined to produce Cathode Copper, a nickel product, and a precious metals slimes product.

For small scale secondary copper operations, the production of black copper can be economically viable. However, the converting and refining of black copper on a smaller scale, is often not economically viable due to the capital costs of the additional plant. Instead, on a smaller scale, the black copper can be exported to an existing primary copper smelter (for processing in Peirce–Smith converters), or leached, using a hydrometallurgical process, to recover the metal values. These two options will not be explored further in this paper.

Aside from the ISASMELT™ furnace, a range of alternative furnace technologies have been used to smelt E-Scrap and other low-grade secondary materials. In the reductive smelting stage, blast furnaces, electric furnaces, and top-blown rotary converters (TBRC) have been used [6,18]. In the converting stage, both Peirce—Smith Converters (PSCs) and TBRCs have been proven.

## 3. Reductive Smelting to Black Copper

The key metallurgical aim of reductive smelting of secondary copper feeds is to minimise the base metals in the 'clean' discard slag while maintaining a fluid slag and molten copper phase. To minimise the base metals in the discard slag, and in turn maximise the base metal distribution to the metal phase, the most reducing conditions practical are required.

Early research [19,20] into the smelting of E-Scrap showed that under extremely reducing conditions, a solid iron phase was formed. In this research, E-Scrap was melted in an electric furnace without modifying the feed or controlling the oxidative potential ($p_{O2}$) in the furnace. As such, the metallic iron and aluminium in the feed were simply melted and reported to a metal phase. Under these conditions, it was observed that the metal products were difficult to handle (not fluid) at temperatures of 1400 °C. In the binary Cu–Fe system (see Figure 1), iron has a limited solubility in the liquid copper phase at secondary copper smelting temperatures. The iron forms a solid phase once the saturation point is reached, represented by the shaded section in Figure 1. As such, the $Fe_{(s)}/FeO_{(l)}$ reaction couple provides the lower $p_{O2}$ limit for reductive smelting, as $FeO_{(l)}$ from the slag starts to reduce to $Fe_{(s)}$ at lower $p_{O2}$ levels. In industrial furnaces, reductive smelting should be controlled and kept as close to this limit as possible with the chosen technology.

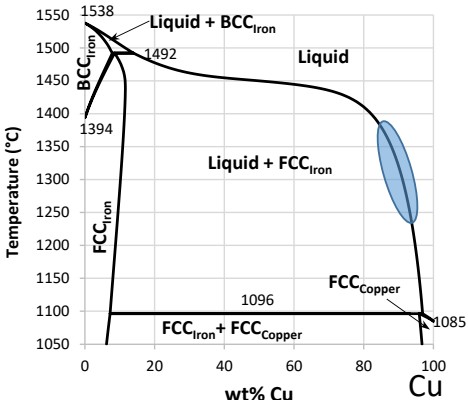

**Figure 1.** Binary Phase Diagram of the Cu–Fe System with the copper liquidus (Fe-FCC primary phase field) temperature highlighted in blue. Calculated using FactSage 8.2 and model parameters [21].

While most of the feed to a secondary copper smelter is low in sulphur, some feeds may be sulphur-rich. If the sulphur in the feed is too high, a matte phase will form at target conditions in the reductive smelting stage. While the matte can be processed in the smelter, it increases the complexity of the flowsheet and requires additional sulphur capture/treatment equipment in the off-gas system. This is illustrated by the Cu–Fe–S ternary isothermal section at 1200 °C shown in Figure 2. In the ternary system, a three-phase field with $Fe_{(s)}$, $Cu_{(l)}$, and matte exists. In this three-phase field, [22] it has been estimated that liquid copper contains 11 wt% Fe and 2 wt% S. In industrial black copper, there are appreciable concentrations of other base metals (Zn, Sn, Pb, etc.) and these impact the composition of the $Fe_{(S)}$, $Cu_{(l)}$, and matte under equilibrium conditions. Further research is required on this more complex system.

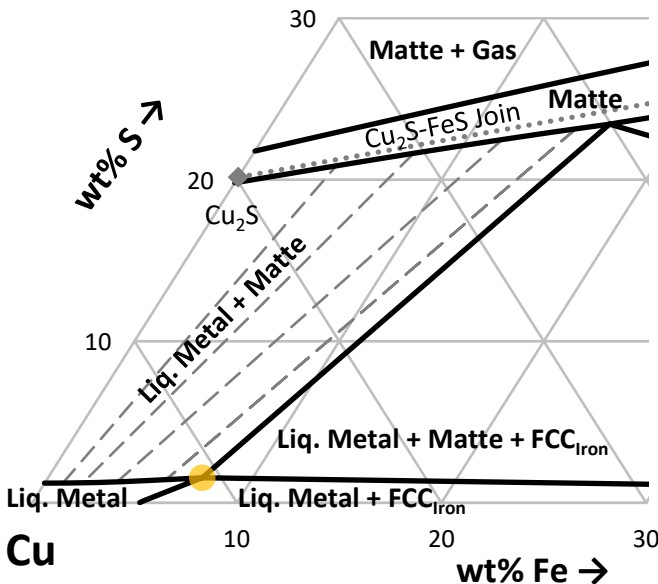

**Figure 2.** Isothermal Section of the Cu–Fe–S System at 1200 °C with the $Cu_{(l)}$–$Fe_{(s)}$–Matte limit indicated by the highlighted orange circle. Calculated using FactSage 8.2 and model parameters [23].

The reductive smelting furnace needs to operate in such a way that the concentration of sulphur in the black copper phase is lower than 2 wt% S, to avoid the formation of a matte phase. While the concentration of iron in black copper can be controlled by adjusting the addition of air or oxygen and reductants, the sulphur content (at typical reductive smelting $p_{O2}$ levels) can only be controlled through feed blending techniques.

The refractory oxides, along with iron and aluminium in the secondary copper scrap, are oxidised and report to the slag phase in the reductive smelting stage. The quantities of these components vary significantly with the secondary scrap feed material. These oxides are fluxed to produce a fluid slag, with the iron-silica (fayalite) based system being the dominant slag type in the secondary copper industry [24]. The major elements of concern are aluminium, zinc, and lime. These can require both iron and silica fluxing to decrease the slag liquidus and position the slag in the desired spinel primary phase field.

A summary of secondary copper slags produced in alternative furnace technologies, and reported in the literature, are given in Table 2. As observed, high alumina and lime concentrations and a wide range of $Fe/SiO_2$ ratios (wt/wt) have been observed to be operable in these operations. In general, it is observed that the TBRC operations are over-fluxed when compared to the Blast Furnace operations. Although not reported explicitly, it is expected that similar slag systems would be operable when smelting E-Scrap rather than 'secondary copper' materials.

**Table 2.** Survey of Slags produced during Reductive Smelting of Secondary Copper Feeds to make Black Copper.

| Smelter | HK AG Blast Furnace [7] | USBM Blast Furnace [17] | USMRC Blast Furnace [6] | James Bri. Blast Furnace [24] | James Bri. Reverb [24] | Brixlegg Blast Furnace [25] | Metallo TBRC [26] | Chemet Co TBRC [27] |
|---|---|---|---|---|---|---|---|---|
| Fe (wt%) | 19.5–35 | 22.5 | 28–36 | 31.6 | 31 | 25.5 | 40.5 | 36 |
| SiO$_2$ (wt%) | 15–25 | 39 | 24–32 | 20 | 19.9 | 28.0 | 20.5 | 14.5 |
| Al$_2$O$_3$ (wt%) | 8–12 | - | 2–11 | 11.7 | 12.2 | 8.0 | - | 5.0 |
| CaO (wt%) | 10–20 | 19 | - | - | - | 6.0 | 8.9 | 2.0 |
| Zn (wt%) | 3–8 | 10 | 5–9 | 6.6 | 9.4 | 6.0 | 8.0 | 10.6 |

To understand the impact of zinc and alumina on the slag liquidus temperature, thermodynamic simulations were performed in FactSage 8.1, using the "*FTOxid*" thermodynamic solution model for Slag and Spinel [28]. A base slag with a Fe/SiO$_2$ ratio of 1.1 wt/wt and a Zn/(Fe+SiO$_2$) ratio of 0.10 wt/wt and an Al$_2$O$_3$/(Fe+SiO$_2$) ratio of 0.17 wt/wt at an equilibrium p$_{O2}$ of $10^{-11}$ atm was used in the simulations. The results are graphed in Figure 3 below. It was determined that a slag with this composition is in the spinel primary phase field. Increases in the concentration of Zn and/or Al$_2$O$_3$ in the slag, generally cause the liquidus temperatures to increase. However, when the concentration of Al$_2$O$_3$ in the slag is less than 9 wt%, the slag moves into the olivine primary phase field and the liquidus temperature increases with decreasing Al$_2$O$_3$.

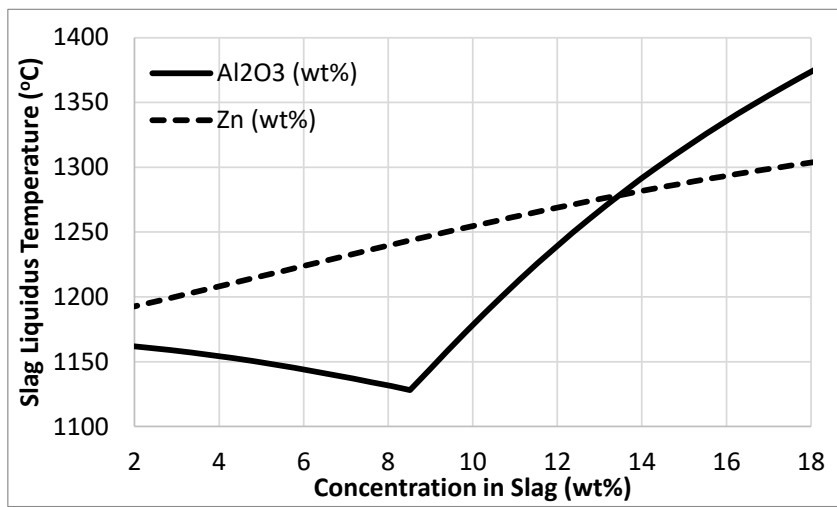

**Figure 3.** The impact of Al$_2$O$_3$ and ZnO on the slag liquidus in the Fe-Si-Al-Zn-O system at a p$_{O2}$ of $10^{-11}$ atm and with a Fe/SiO$_2$ ratio of 1.1 wt/wt, Zn/(Fe+SiO$_2$) ratio of 0.10 wt/wt, Al$_2$O$_3$/(Fe+SiO$_2$) ratio of 0.17 wt/wt [28].

In the secondary smelting process, the removal of zinc is typically challenging. Zinc has a distribution ratio close to 1.0 during the reductive smelting stage [7,29]. As such, the zinc commonly contaminates the discard slag from secondary copper smelters. This is partially managed by fuming the zinc as a metal vapour species Zn$_{(g)}$, but this is difficult to achieve with both Blast Furnace and TBRC technology, as evident by the slag compositions reported in the literature, as seen in Table 2.

A comparison of the fuming rates for alternative secondary copper smelting furnaces is given in Table 3.

**Table 3.** Fuming of Elements in Secondary Copper Furnaces [25,26].

| Element | Blast Furnace | TBRC |
|---|---|---|
| Sn | 2.4% | 17% |
| Pb | 37% | 25% |
| Zn | 31% | 46% |

A preliminary simulation of the possible zinc fuming rates during E-Scrap smelting was performed with FactSage 8.0, using the "*FTOxid*" solution model for the slag and spinel phase, and "*FTMisc*" for the molten copper liquid phase [28]. The smelting of the E-Scrap in Table 1 Ref. 4 [14] was simulated at a $p_{O2}$ of $10^{-11}$ atm and at a temperature of 1250 °C. The equilibrium zinc concentration in the slag was found to be less than 0.5 wt%, significantly lower than the slag concentrations reported in Table 2.

## 4. Converting Black Copper to Raw Copper

The black copper produced during reductive smelting contains an appreciable quantity of zinc, lead, tin, nickel, and iron. These species detrimentally impact the electrolytic refining process and are removed in the converting stage. The zinc, lead, tin, and iron can be reduced to low levels, but only a portion of the nickel can be removed. Unlike primary copper converting, the batch converting of black copper may not generate sufficient energy for autogenous processing and fuel may be required. The composition of black copper produced in industrial secondary copper smelters is provided in Table 4.

**Table 4.** Survey of Black Copper from Reductive Smelting to Black Copper.

| Smelter | HK AG Blast Furnace [7] | USBM Blast Furnace [17] | USMRC Blast Furnace [6] | Brixlegg Blast Furnace [25] | Metallo TBRC [26] |
|---|---|---|---|---|---|
| Cu (wt%) | 72–80 | 75–88 | 75–80 | 80 | 78.3 |
| Fe (wt%) | 4–6 | 0.5–1.5 | 4–6 | 5 | 3.5 |
| Ni (wt%) | 2–4 | - | 4–6 | 4 | 2.9 |
| Sn (wt%) | 3–8 | 1.5 | 4–6 | 3.5 | 3.4 |
| Pb (wt%) | 2.5–6 | 1.5 | 4–6 | 2 | 4.3 |
| Zn (wt%) | 3–8 | 4–10 | 4–6 | 4 | 6.6 |

There are four ways in which these elements are removed during converting:

- Oxide/Metal Fuming
- Selective oxidation using a fayalitic slag
- Sulphide fuming
- Selective oxidation using a calcium ferrite slag

The oxide/metal fuming process fumes tin, zinc, and lead as oxide and metal fumes, and partitions the iron and other oxides into a silica slag phase. To perform this process, the converter is run hot, with the addition of coal or coke to fume off the tin (which is vaporised as $SnO_{(g)}$ under reducing conditions) [25,30]. For maximum fuming efficiency, the iron is first oxidised and the fayalite slag that is produced is skimmed out, prior to the fuming of the tin. This process provides a tin-, zinc-, and lead-rich fume which can be further processed to produce a saleable by-product. The major advantage of this process over the other converting options is that the Cu/Sn (wt/wt) ratio in the fume is low, simplifying the processing of the fume. However, the removal of tin with this process is not efficient and this leads to high recirculating loads of tin and other base metals in the smelter.

The selective oxidation route, using a fayalitic slag, is used as an alternative to the oxide fuming process. In this process, the impurities are oxidised into a fayalitic slag with

the addition of a silica flux to maintain a fluid slag. It was found that a two-stage slagging process was most effective, with an iron- and zinc-rich slag formed first, followed by a tin, lead-, nickel-, and copper-rich slag in the second stage [6,30]. The first stage slag can be recycled back to the reductive smelting stage and the second stage slag is processed separately so base metals can be recovered. This process has the advantage of lower coal consumption compared to the oxide fuming process. However, the second stage slag has a high Cu/Sn (wt/wt) ratio leading to a more complex reprocessing route for this converter slag when compared to the tin-rich dusts.

An alternative to these two historical processes is a sulphide fuming process, which relies on the high vapour pressure of $SnS_{(g)}$ and $PbS_{(g)}$ to fume these species. To perform a sulphide fuming process, a sulphide material (e.g., sulphur or pyrite) could be added to the converter prior/during fuming whilst maintaining reducing conditions (e.g., via addition of coal or coke). Alternatively, the black copper could be charged into a Peirce–Smith converter along with copper matte from a primary copper smelter.

When the black copper and matte are first mixed, a separate matte and copper phase are formed as there is insufficient sulphur present to 'matte' all of the copper phase. In the first stage of converting, the iron in the matte is oxidised to form a fayalitic slag and little $SO_{2(g)}$ is generated. This stage occurs in highly reducing conditions and most of the zinc is fumed (as a $Zn_{(g)}$ species). This stage ends relatively quickly, once sufficient iron has been oxidised to incorporate all the black copper into the copper matte phase. The overall chemical reaction for this stage is:

$$4\,Cu_{(l,metal)} + 2\,FeS_{(l,matte)} + O_{2(g)} \rightarrow 2\,Cu_2S_{(l,matte)} + 2\,FeO_{(l,slag)}$$

The second and third stages of converting which follow the conventional batch copper converting process are slag blow and blister blow. The tin and lead are fumed faster at the start of the slag blow. In addition, the tin and lead mostly report to the fayalitic slag that is formed during the converting process. The combined converting of black copper and primary copper matte has the advantages of utilising existing smelting equipment and entailing low fuel consumption. It does, however, remove some converting capacity from the primary smelter. A sulphide fuming process would also produce dust and slag that:

1.  Should be kept separate from other primary smelter converting dust/slags (where possible);
2.  Requires further processing to recover the tin and lead from slag.

An alternative process would be to use a lime-based slag, as practiced in the ISACON-VERT™ process [31]. In this process, the base metals would be oxidised to form a calcium–ferrite slag. The slag is considered a 'basic slag', while the traditional fayalitic slag is an 'acidic slag'. A basic slag would enable a higher rejection and partitioning of some base metals. Further research on the use of this slag for black copper converting is required.

## 5. Refining of Raw/Blister Copper to Produce Cathode Copper and PM Sludge

At the conclusion of black copper converting, a raw or blister copper is produced. This copper still contains some impurities and a high concentration of oxygen dissolved as $Cu_2O_{(l)}$ [25]. Fire refining is performed to remove these impurities and the oxygen in the raw copper. Once the oxygen has been removed, the copper is considered anode copper and is cast into anodes for Electro-Refining.

During electrolytic refining, the nickel is oxidised from the anodes and reports to the electrolyte, with the electrolyte being processed to produce a nickel by-product [32]. Other metals report to the anode slimes, along with the precious metals. These slimes are subsequently treated in a multi-stage process to recover the gold, silver, and precious metals.

## 6. The ISASMELT™ Process for Reductive Smelting of E-Scrap

E-Scrap can be smelted in an ISASMELT™ furnace by itself or combined with other copper bearing feeds, such as concentrates. When processed with other copper feeds, the

E-Scrap is pneumatically injected down the lance while the other feed materials are fed into the furnace via the final feed device and feed chute. The ISASMELT™ furnace in this situation would be operated at the conditions required by the copper-bearing feed. In contrast, a dedicated E-Scrap ISASMELT™ furnace presents numerous technical challenges due to the unique feed materials.

The major challenges associated with smelting E-Scrap, when compared to other secondary copper feeds, are the low concentrations of copper, the high levels of plastic, and the high aluminium content. This results in a lower copper/slag ratio in the furnace, significant quantities of energy being generated in the furnace, the potential of dioxan and furan generation, and the potential need for simultaneous iron and silica fluxing. The ISASMELT™ technology has several unique features to manage these challenges.

The feed to an E-Scrap ISASMELT™ furnace consists of E-Scrap, iron flux, silica flux, coolant, and potentially other recycles (residues, dusts, scrap). The E-Scrap is shredded, as required for feed sampling, without PCB disassembly. In the ideal situation, a primary smelting slag is used as the iron source as it is usually a very low-cost material containing both iron and silica (but typically more iron). This slag would be either crushed or granulated and fed onto the final feed device before entering the furnace via the feed port. The silica flux would be a low-cost siliceous material obtained locally, as low in alumina as possible, also fed onto the final feed device. The E-Scrap would first be shredded and sampled prior to blending, and then pneumatically injected into the ISASMELT™ bath via the submerged lance. The E-Scrap is injected directly into the bath to minimise carryover of high value fines into the off-gas and to ensure that all the organic material enters the bath for destruction at a high temperature. In addition, due to the high energy content of the feed materials, coolants are added to the furnace to manage the temperature within the bath. Both copper scrap and recycled E-Scrap ISASMELT™ slag can be used for this.

The E-Scrap is injected into the bath, along with air and oxygen. The organics in the E-Scrap are destroyed in the molten bath, which is critical to controlling the formation of the dioxins and furans that are of significant concern when processing halide-containing plastics. The iron and aluminium in the feed are oxidised and report to the slag phase. The zinc is fumed, along with a portion of the lead in the feed. The copper reports to the black copper phase, along with the remaining base and precious metals. The slag is batch tapped from the furnace via the slag taphole and the black copper is batch tapped from the furnace via a separate copper taphole. The slag is subsequently settled for the recovery of entrained copper, prior to sale (as a clean, safe, and inert material) or discarding.

One of the major challenges discussed in regard to smelting E-Scrap in alternative technologies, such as Blast Furnaces or TBRC, is the difficulty in fuming zinc effectively. The intense agitation of the ISASMELT™ bath, coupled with the direct contact of the gas with the slag, typically results in very high rates of fuming. As shown in the FactSage slag liquidus calculations, a significant reduction of zinc in the slag does not only improve the recovery of metals but also decreases the slag viscosity, hence reducing the operating temperature and fluxing requirements of the smelting process.

The other major challenge in smelting E-Scrap with conventional technology, particularly TBRC technology, is the formation of an operable slag. As observed in the slags reported for secondary copper TBRC sites, significant fluxing is performed to dilute the concentration of alumina ($Al_2O_3$) and lime ($CaO$). Both components stabilise the formation of solids, with alumina stabilising the formation of spinel and lime stabilising a range of phases depending on the primary phase field of the slag. Due to the top-blown nature of the TBRC process, the slag must be operated with a sufficiently high superheat to prevent accretion formation in the furnace.

The intense agitation of the ISASMELT™ bath enables the slag to be held at a lower thermal set point when compared to other smelting furnace technologies. Continuous operation in a stationary vessel allows for the use of advanced temperature and slag chemistry control systems. Combined, these two points allow the ISASMELT™ technol-

ogy to operate with lower flux additions compared to conventional secondary copper smelting technology.

The bath gases generated in the E-Scrap ISASMELT™ furnace contain high levels of $CO_{(g)}$ and halides, along with the zinc and lead fume. This off-gas is processed in several stages to treat and capture these species prior to discharging a clean gas stream into the atmosphere. The off-gas is extracted from the furnace and undergoes afterburning in a radiative waste heat boiler (WHB). In the WHB, the $CO_{(g)}$ and metal fume is combusted, generating heat. The heat is extracted, and the gas is partially cooled, heating the steam in the WHB. Importantly, the WHB provides sufficient temperature, mixing, and residence time for the destruction of any remaining hydrocarbons, dioxins, and furans. The steam generated in the waste heat boiler can be used for heating elsewhere on site, or to generate power in a steam turbine. After post combustion, the gas is quench cooled and fine dust is removed. The gas is finally passed through a scrubber, removing the HCl and HF in the off-gas.

The sealed ports on the ISASMELT™ technology, and the use of advanced off-gas systems, enables the technology to operate in a clean and sustainable manner. The ISASMELT™ technology has been proven to meet the most stringent European environmental regulations [33].

Due to the unique characteristics of the E-Scrap ISASMELT™ furnace, the slag quality requirements and fuming kinetics are considered superior to alternative E-Scrap smelting technologies. Piloting of this technology was identified as critical to understanding the slag fluxing requirements and zinc fuming performance.

## 7. ISASMELT™ Pilot Plant Description

E-Scrap ISASMELT™ furnace piloting was performed to understand the slag fluxing window and optimisation for an industrial scale furnace. The pilot plant facilities at the CSIRO laboratories at Clayton, Victoria, Australia were used for the pilot testing—details of this facility are covered in a separate paper [34].

Prior to use in the pilot plant, the feed materials were crushed (where required) using a laboratory jaw crusher and screened using conventional screening pans. The materials used in the pilot plant were fayalite slag, silica flux, iron flux, coal, high-grade copper scrap, and E-Scrap. Assays for the flux and fuel materials used at the pilot plant are provided in Tables 5 and 6.

**Table 5.** Fluxes used in the E-Scrap ISASMELT™ Pilot Testing.

| Material | $Al_2O_3$ | CaO | Fe | MgO | $SiO_2$ |
|---|---|---|---|---|---|
| Fayalite Slag | 2.53 | 3.27 | 38.0 | 2.28 | 32.3 |
| Silica Flux | 1.03 | - | 0.30 | 0.09 | 97.6 |
| Iron Flux | 1.88 | - | 63.7 | 0.04 | 3.62 |

**Table 6.** Fuel used in the E-Scrap ISASMELT™ Pilot Testing.

| Sample | Moisture (%) | Ash (Dry %) | Volatiles (Dry %) | Fixed Carbon (Dry %) | C (Dry %) | H (Dry %) | N (Dry %) | S (Dry %) | Other (Dry %) |
|---|---|---|---|---|---|---|---|---|---|
| Coarse Coal | 6.64 | 2.23 | 29.3 | 0.899 | 79.2 | 4.3 | 1.82 | 0.2 | 14.5 |

Pre-shredded lots of E-Scrap were smelted, each being 80 to 120 kg. The E-Scrap was weighed out, along with a preliminary flux blend, and agglomerated, ready for smelting.

The pilot furnace, as illustrated in Figure 4, was heated to the target smelting temperature prior to building a starting bath. The E-Scrap was then smelted using air and oxygen down the ISASMELT™ lance.

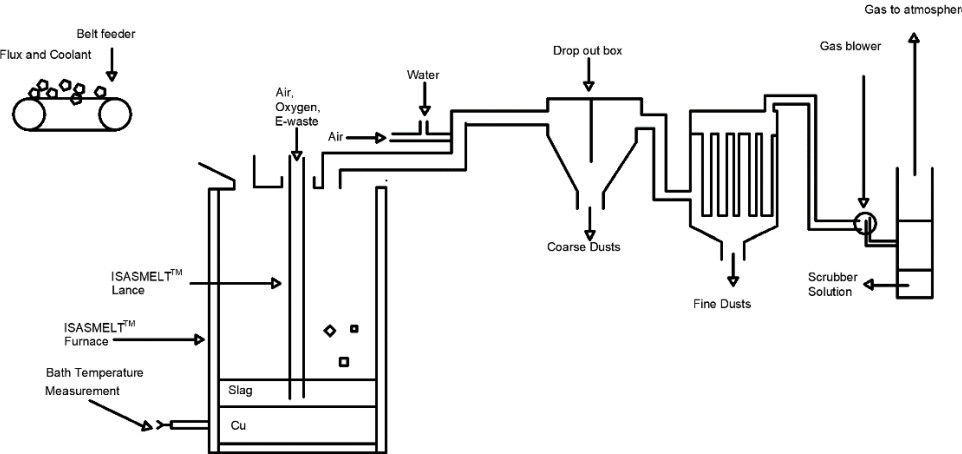

**Figure 4.** Pilot Plant Set-Up [34].

The process off-gases from the pilot furnace were first post-combusted, and then cooled via radiative heat loss and evaporative spray cooling (as required). This was followed by dust collection in a baghouse and cleaning by a wet scrubber.

The slag and black copper were sampled every 20–30 min during piloting.

When all feed panels were smelted, the lance was raised out of the furnace to perform a holding burner function and the molten materials were removed into cast iron pots.

## 8. ISASMELT™ Pilot Plant Trials of E-Scrap Smelting

A range of mixed E-Scrap feeds were smelted to test the flexibility of the process.

The key aims of the pilot was to determine the extent of zinc fuming and the window of operable slag conditions in the furnace.

Preliminary simulations were performed in GT's proprietary model to determine an estimate of operable furnace starting conditions.

To determine the extent of zinc fuming, the E-Scrap trials were performed close to iron saturation in the black copper, maximising the equilibrium partial pressure of $Zn_{(g)}$ under the reductive smelting conditions. Based on the previous discussion (Figure 1), it was expected that this would be approximately 2–4 wt% Fe in the black copper. During smelting, the air/E-Scrap ratio was adjusted to achieve the target iron in the black copper.

To understand the range of operable slags, the fluxing was adjusted while smelting—based on the observed slag fluidity during smelting. The fluxing/E-Scrap ratios required in the furnace were determined based on the lance operation (during smelting) and features of the slag (during sampling). An initial $Fe/SiO_2$ ratio of 0.6 to 1.8 wt/wt was targeted, with different $Fe/SiO_2$ ratios observed during the tests. The tests performed also helped to understand the maximum alumina ($Al_2O_3$) operable in the pilot ISASMELT™ furnace.

## 9. ISASMELT™ Pilot Plant Results from E-Scrap Smelting

All pilot plant tests were successful, with a clean molten slag and a black copper product produced. It was found that operating the furnace at close to iron saturation resulted in stable conditions. Variations in the furnace operating temperature (+/−75 °C) and $p_{O2}$ (<0.1 wt% Fe in black copper, through to iron formation) were experienced. It was demonstrated that the slag and operating conditions could be adjusted to compensate so that the furnace was still operable within such wide bath temperatures and $p_{O2}$ variations.

Dips of fluid slags were taken during operation, with over 50 operable slag samples collected. A wide range of $Fe/SiO_2$ ratios were found to be operable. The range of slag compositions is provided in Table 7. In addition, it was found that slags with 20 wt% $Al_2O_3$ were operable, significantly higher than other furnace technologies. Further research is required in the Fe–Si–Al–O slag system and the impact of minor elements on the phase equilibria and slag physio-chemical properties.

**Table 7.** Operable Slags in the E-Scrap ISASMELT™ Furnace.

| | Concentration (wt%) |
|---|---|
| Fe (wt%) | 15–35 |
| SiO$_2$ (wt%) | 10–35 |
| Al$_2$O$_3$ (wt%) | 5–20 |

The slag, black copper, and dusts from the pilot trials were collected, weighed, sampled, and assayed to perform a detailed mass balance on the E-Scrap ISASMELT™ furnace. A summary of the weights of the phases, and the concentration of critical metals—zinc, tin, nickel, and lead—are provided in Table 8.

**Table 8.** Slag, Copper, and Dust assays from the E-Scrap ISASMELT™ pilot testing.

| Component | Slag | Black Copper | Coarse Dust | Fine Dust |
|---|---|---|---|---|
| Sn (wt%) | 0.43 | 4.58 | 1.76 | 7.46 |
| Ni (wt%) | 0.07 | 0.69 | 0.11 | 0.03 |
| Pb (wt%) | 0.05 | 0.22 | 1.59 | 9.99 |
| Zn (wt%) | 0.22 | 0.04 | 4.74 | 13.8 |
| Mass (kg) | 159 | 44 | 1.20 | 10.9 |

The deportment of the critical metals was calculated based on a mass balance from the weights and concentrations presented in Table 8. The deportment of the tin, lead, nickel, and zinc is reported in Table 9.

**Table 9.** Element Distribution in the E-Scrap ISASMELT™ furnace.

| Element | Slag | Metal | Fume |
|---|---|---|---|
| Sn | 19% | 57% | 24% |
| Ni | 27% | 72% | 1% |
| Pb | 6% | 7% | 87% |
| Zn | 18% | 1% | 81% |

It was found that most of the lead and zinc reported to the fine dust. Based on the distributions, it was shown that the intense ISASMELT™ bath agitation results in fuming rates that are unobtainable with alternative furnace technologies. As a result of the high fuming rates, a discard slag with very low concentrations of lead and zinc was produced.

When treating similar feeds, the performance of the ISASMELT™ pilot plant compared to other furnace technologies, demonstrates that reaction kinetics and high-temperature fluid properties are significant factors in the success of industrial processes. Further fundamental research is required on phase equilibria and minor element distribution for both the molten black copper and slag phases relevant to E-Scrap and secondary copper smelting.

## 10. Conclusions

The smelting of E-Scrap can be performed using the conventional secondary copper smelting flowsheet. While the E-Scrap feed differs from other secondary copper materials, the successful use of a similar slag system has been demonstrated. Of the three stages of smelting, reductive smelting to producing black copper has the most challenging process chemistry. Other secondary copper smelting furnace technologies, such as the Blast Furnace and Top Blown Rotary Converters (TBRC), are unable to fume appreciable quantities of

zinc due to kinetic limitations. In addition, due to the furnace technology, over-fluxing is typically performed to produce a fluid slag.

The E-Scrap ISASMELT™ furnace was piloted, demonstrating high rates of zinc fuming and the ability to operate with high alumina concentrations in the slag. The ISASMELT™ technology was shown to produce a slag with low concentrations of lead, zinc, and tin. Further research on the kinetics of zinc fuming and slag thermodynamics relevant to E-Scrap smelting is required. It was demonstrated that the E-Scrap ISAS-MELT™ overcomes the major challenges reported when smelting E-Scrap using alternative furnace technologies.

The continuously-fed operating mode of the E-Scrap ISASMELT™ furnace allows the slag chemistry and bath temperature to be controlled within a tight window. This means that the chemical, and particularly mechanical, entrainment of copper in the discard slag from an E-Scrap ISASMELT™ is significantly lower than with alternative technologies. This has a significant impact on the treatment/use/disposal of this slag and (perhaps more importantly) the overall recovery of copper and precious metals.

**Author Contributions:** Conceptualization, S.N. (Stuart Nicol), B.H., O.M. and S.N. (Stanko Nikolic); methodology, B.H. and O.M formal analysis, S.N. (Stuart Nicol); investigation, S.N. (Stuart Nicol), B.H., O.M. and S.N. (Stanko Nikolic); resources, B.H. and O.M; data curation, B.H..; writ-ing—original draft preparation, S.N. (Stuart Nicol); writing—review and editing, B.H. and S.N. (Stuart Nicol); supervision, S.N. (Stanko Nikolic); project administration, B.H.; funding acquisition, B.H. All authors have read and agreed to the published version of the manuscript.

**Funding:** This research received no external funding.

**Data Availability Statement:** Not applicable.

**Acknowledgments:** The authors, and Glencore Technology, would like to thank CSIRO for the use of their SiroSmelt pilot plant facilities. In particular, the authors appreciate the assistance from Michael Somerville in the operation of the pilot furnace. The authors would also like to thank Denis Shishin for assistance in preparing Figures 1 and 2 The authors would also like to thank Shuwenjun (Winnie) Ma for assistance in preparing Figure 3.

**Conflicts of Interest:** The authors declare no conflict of interest.

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
