# Peer review of "Extraction and Recovery of Critical Metals from Electronic Waste Using ISASMELT™ Technology"

_processes, doi:10.3390/pr11041012_

Round 1

Reviewer 1 Report

The study about application of ISASMELT™ Technology for recycling of e-waste is very interesting and is well drafted. Here are some minor comments and suggestions to make this work more sensible for a wider range of audiences.

1.      In Abstract, it is suggested to add a few details about the results if possible. For examples, what type of fluxes were studied, what concentration is considered high level for zinc in fume, what concentration of alumina is considered high in the slag, what concentration of zinc in the slag is considered low? maybe just mention some quantities in parenthesis.

2.      In Introduction, second paragraph, where the authors are mentioning the two challenges of e-waste smelting, it is recommended to add one sentence explaining about how high level of alumina is affecting the properties of slag. (Also, citation “A. M. Mostaghel 2013” is typed twice, please remove one of them).

3.      In introduction, where the ISASMELT™ technology is being described, please add a brief explanation about the way this technology operates. (I understand it is explained in later subsections).

4.      In the last line of introduction, it is suggested to mention the topics that are covered in the following sections of the paper.

5.      There are more recent references about pyrometallurgy of e-waste recycling, including technologies, slag adjustment, effect of fluxes, emission control, etc. Perhaps there could be some mentions about the recent consideration and directions in pyrometallurgy of e-waste.

6.      Please mention a little about the plastic parts of e-waste. Are they removed from the feed prior to smelting? Are they considered as a role player in the reduction reactions? If they burn at temperatures lower than 800 C, they can generate dangerous fumes (probably this is not the case here but could be mentioned).

7.      What is the range of particle size for the feed? Are they pretreated for the plastics and ceramic parts?

8.      What about the disadvantages and areas the technology may not be the best option and should be improved?

Author Response

Reviewer 1

The study about application of ISASMELT™ Technology for recycling of e-waste is very interesting and is well drafted. Here are some minor comments and suggestions to make this work more sensible for a wider range of audiences.

  1. In Abstract, it is suggested to add a few details about the results if possible. For examples, what type of fluxes were studied, what concentration is considered high level for zinc in fume, what concentration of alumina is considered high in the slag, what concentration of zinc in the slag is considered low? maybe just mention some quantities in parenthesis.

The author thanks the reviewer for identifying this. The sentence in the abstract now reads “The testing demonstrated that high levels of zinc fuming (>80%) are obtained in the E-Scrap ISASMELT™ furnace, decreasing the iron and silica flux additions required to manage the detrimental viscosity effects of zinc in the slag. In addition, it was demonstrated that slags containing high concentrations of alumina (>10 wt%) are operable in an ISASMELT™ furnace.”

  1. In Introduction, second paragraph, where the authors are mentioning the two challenges of e-waste smelting, it is recommended to add one sentence explaining about how high level of alumina is affecting the properties of slag. (Also, citation “A. M. Mostaghel 2013” is typed twice, please remove one of them).

The second paragraph has been modified such that the following sentence has been included “The alumina and zinc in slag both stabilise the spinel phase, which increases the slag liquidus temperature and results in either higher operating temperatures or a higher slag viscosity (A. M. Mostaghel 2013) (A. M. Mostaghel 2013) (Klemettin 2017).”

“A. M. Mostaghel 2013” refers to two different papers. These are now identified as “a” and “b”.

  1. In introduction, where the ISASMELT™ technology is being described, please add a brief explanation about the way this technology operates. (I understand it is explained in later subsections).

The author thanks the reviewer for identifying this. The following has been added to the introduction “The ISASMELT™ furnace is a bath smelting furnace, with the ISASMELT™ lance submerged in the slag layer and injecting reaction gases directly into the bath. In this furnace, the slag phase is used to carry oxygen to the feed materials suspended in the bath.”

  1. In the last line of introduction, it is suggested to mention the topics that are covered in the following sections of the paper.

The author thanks the reviewer for identifying this. The following has been added to the introduction “The secondary copper process chemistry and slag design will be reviewed to understand E-Scrap Smelting. The pilot results for an E-Scrap ISASMELT™ furnace will be discussed and examined.”

  1. 5.There are more recent references about pyrometallurgy of e-waste recycling, including technologies, slag adjustment, effect of fluxes, emission control, etc. Perhaps there could be some mentions about the recent consideration and directions in pyrometallurgy of e-waste.

The author thanks the reviewer for identifying this. Two recent reviews of E-Scrap processing have been identified and referenced “Reviews of E-Scrap have been performed, outlining the processes and challenges when smelting this material (Zhang, L. and Xu, Z., 2016) and (Cui, J. and Zhang, L., 2008).” Due to the amount of research in this field, readers have been directed to these review papers

  1. Please mention a little about the plastic parts of e-waste. Are they removed from the feed prior to smelting? Are they considered as a role player in the reduction reactions? If they burn at temperatures lower than 800 C, they can generate dangerous fumes (probably this is not the case here but could be mentioned).

The author thanks the reviewer for identifying this. Two sentences have been modified to address this, 268-270, “This results in a lower copper/slag ratio in the furnace, significant quantities of energy being generated in the furnace, the potential of dioxan and furan generation and the potential need for simultaneous iron and silica fluxing.” and 280-282 “The E-Scrap is injected directly into the bath to minimise carryover of high value fines into the off gas and to ensure that all the organic material enters the bath for destruction at high temperature.“

  1. What is the range of particle size for the feed? Are they pre-treated for the plastics and ceramic parts?

The E-Scrap pre-treatment has been specified in lines 273-274 “The E-Scrap is shredded, as required for feed sampling, without PCB disassembly.”

  1. What about the disadvantages and areas the technology may not be the best option and should be improved?

The authors are currently unaware of any disadvantages in the ISASMELT™ furnace technology. Significant research and development has been performed over the last 30 years to address any issues which were identified in the first generation ISASMELT™ technology. If any disadvantages are identified going further, R&D will be performed to address these.

Reviewer 2 Report

……………………….of life electronics (E-Scrap) need to be responsibly………… correct as

…………..of life electronics equipment (E-Scrap) need to be responsibly……………

What are Critical metals?

The composition of E-Scrap varies significantly, changing with the electronic good, 35

the manufacturer, and the year of manufacture. Need rewriting.

(A. M. Mostaghel 2013) (A. M. Mostaghel 2013) (Klemettin 2017)) correct this.

Table 6 Why volatile S is so high?

How stoichiometry of all components in coarse coal is matching?

While the E-Scrap feed differs from other secondary copper materials, the same slag system can be used. Not clear.

Give maximum references from the last 15-20 years and only very essential ones before 2000.

Reference cited need to be in one pattern only (Asper the format of the journal). 

Author Response

……………………….of life electronics (E-Scrap) need to be responsibly………… correct as

 …………..of life electronics equipment (E-Scrap) need to be responsibly……………

The author thanks the reviewer for finding this mistake. The sentence now reads “The processing of end-of-life electronics equipment (E-Scrap) is becoming increasingly important to maintain the supply of the critical metals required globally, and to reduce environmental pollution.”

What are Critical metals?

For this special issue, the critical metals have been defined as “Ferroalloy metals: chromium, cobalt, manganese, nickel, niobium, tantalum, titanium, and vanadium; Precious metals: gold, silver, and platinum group metals (iridium, palladium, platinum, rhodium, and ruthenium); Other non-ferrous metals: aluminium, antimony, bismuth, cadmium, copper, indium, gallium, germanium, lead, lithium, and magnesium. In our introduction, we have highlighted which of these listed metals are in E-Scrap.

The composition of E-Scrap varies significantly, changing with the electronic good, 35

the manufacturer, and the year of manufacture. Need rewriting.

The author thanks the reviewer for finding this mistake. The sentence now reads “The composition of E-Scrap varies significantly, dependent on the electronic good, the manufacturer, and the year of manufacture”

(A. M. Mostaghel 2013) (A. M. Mostaghel 2013) (Klemettin 2017)) correct this.

            The author thanks the reviewer for finding this mistake. The referencing has been corrected.

Table 6 Why volatile S is so high?

This particular coal was not a coking coal, it has a high quantity of volatiles and high sulphur. For this application, this coal was preferred for economic purposes. This is one the advantages of the technology – it can use a range of low-grade solid / liquid/ gaseous fuels.

How stoichiometry of all components in coarse coal is matching?

            Yes, the composition adds up (C + H + N + S + Others =100%)

While the E-Scrap feed differs from other secondary copper materials, the same slag system can be used. Not clear.

The author thanks the reviewer for identifying this. This section now reads. ” While the E-Scrap feed differs from other secondary copper materials, the successful use of a similar slag system has been demonstrated.”

Give maximum references from the last 15-20 years and only very essential ones before 2000.

The author thanks the reviewer for identifying this. Two recent reviews of E-Scrap processing have been identified and referenced “Reviews of E-Scrap have been performed, outlining the processes and challenges when smelting this material (Zhang, L. and Xu, Z., 2016) and (Cui, J. and Zhang, L., 2008).” Due to the amount of research in this field, readers have been directed to these review papers 

Reference cited need to be in one pattern only (Asper the format of the journal). 

The author thanks the reviewer for finding this mistake. The referencing has been corrected.

Reviewer 3 Report

E-wastes are a source of many metals critical to electronics manufacturing, including copper, nickel, tin, zinc and precious metals, but they are also an important source of environmental concern. Therefore, E-wastes must be treated to recover important metals, but also to eliminate an important environmental problem.

The number of references from the last 5 years is small compared to the total number of references. Please, if possible, cite more recent relevant references.

Line 8: The focus can also be the environment.

Table 1: Which E-Scrap did the authors, mentioned in the Table (Shuey, Kim and Iji), study? I think it would be better to mention E-Scrap as well.

Line 46: Would not it be better to insert a space among Tables and text?

Figure 2: What does the orange dot mean? I think Figure 2 should be self-explanatory so that the reader can understand it without too much effort.

Line 124: Mass balance, right?

Line 127: Insert a space between the Figure and the text.

Line 368: I think it would be very interesting if the authors showed a typical reactor feed composition. If possible, the feed composition that generated the results shown in Tables 8 and 9.

Line 397: Was the reaction kinetics studied and not presented in this paper or is it literature data? If it is literature data, please provide the reference.

References: Is it not necessary to cite the volume of the journals in the references?

Author Response

E-wastes are a source of many metals critical to electronics manufacturing, including copper, nickel, tin, zinc, and precious metals, but they are also an important source of environmental concern. Therefore, E-wastes must be treated to recover important metals, but also to eliminate an important environmental problem.

The author thanks the reviewer for finding this mistake. The sentence now reads “The processing of end-of-life electronics equipment (E-Scrap) is becoming increasingly important to maintain the supply of the critical metals required globally, and to reduce environmental pollution.”

The number of references from the last 5 years is small compared to the total number of references. Please, if possible, cite more recent relevant references.

The author thanks the reviewer for identifying this. Two recent reviews of E-Scrap processing have been identified and referenced “Reviews of E-Scrap have been performed, outlining the processes and challenges when smelting this material (Zhang, L. and Xu, Z., 2016) and (Cui, J. and Zhang, L., 2008).” Due to the amount of research in this field, readers have been directed to these review papers

Line 8: The focus can also be the environment.

The text has now been updated to read “The processing of end-of-life electronics is becoming increasingly important to maintain the supply of the critical metals required globally, and to reduce environmental pollution.”

Table 1: Which E-Scrap did the authors, mentioned in the Table (Shuey, Kim and Iji), study? I think it would be better to mention E-Scrap as well.

In this study, an E-Scrap with a different bulk composition was used. However, as part of the tests, mixed and unknown feeds were smelted in order to test flexibility of the process. As such, the bulk composition information is not available. Line 369 updated to reflect this.

Line 46: Would not it be better to insert a space among Tables and text?

We thank the reviewer for finding this mistake. The text has been updated to address this.

Figure 2: What does the orange dot mean? I think Figure 2 should be self-explanatory so that the reader can understand it without too much effort.

The caption has now been updated to “Cu(l)-Fe(s)-Matte limit indicated by highlighted orange circle.” to provide further clarity to the reader.

Line 124: Mass balance, right?

Yes, feed blending is term which includes both mass balances and the practice of mixing materials to achieve a desired bulk composition. However, as ‘mass balance’ is performed for other applications within a smelter, the word ‘feed blending techniques’ is used in this situation to be specific as to the sort of mass balance which is performed.

Line 127: Insert a space between the Figure and the text.

            We thank the reviewer for finding this mistake. The text has been updated to address this.

Line 368: I think it would be very interesting if the authors showed a typical reactor feed composition. If possible, the feed composition that generated the results shown in Tables 8 and 9.

We agree with the reviewer that this information would add value to this paper. However, as part of the tests, mixed and unknown feeds were smelted in order to test flexibility of the process. As such, this information is not available to us. However, the complete mass balance is able to be performed based on the detailed assays and weights of the smelting products. Sufficient information was collected to enable the complete mass balance to be closed.

Line 397: Was the reaction kinetics studied and not presented in this paper or is it literature data? If it is literature data, please provide the reference.

The significant differences in fuming rates compared with other furnace technology, operating under similar conditions, provides evidence that there is kinetic processes and reactions which are impacting the rates of these fuming reactions. The fundamental research required to determine the rates of reactions has not been determined. However, there is still sufficient evidence from the tests performed that the reaction kinetics are critical. Line 416 update to explain this.

References: Is it not necessary to cite the volume of the journals in the references?

The author thanks the reviewer for finding this mistake. The referencing has been corrected.

Round 2

Reviewer 2 Report

 Minor language corrections may be required.